# Knowledge Discovery and Dataset for the Improvement of Digital Literacy Skills in Undergraduate Students

**Pongpon Nilaphruek and Pattama Charoenporn \***

Department of Computer Science, School of Science, King Mongkut's Institute of Technology Ladkrabang, Bangkok 10520, Thailand; 63605014@kmitl.ac.th
\* Correspondence: pattama.ch@kmitl.ac.th

**Abstract:** For over two decades, scholars and practitioners have emphasized the importance of digital literacy, yet the existing datasets are insufficient for establishing learning analytics in Thailand. Learning analytics focuses on gathering and analyzing student data to optimize learning tools and activities to improve students' learning experiences. The main problem is that the ICT skill levels of the youth are rather low in Thailand. To facilitate research in this field, this study has compiled a dataset containing information from the IC3 digital literacy certification delivered at the Rajamangala University of Technology Thanyaburi (RMUTT) in Thailand between 2016 and 2023. This dataset is unique since it includes demographic and academic records about undergraduate students. The dataset was collected and underwent a preparation process, including data cleansing, anonymization, and release. This data enables the examination of student learning outcomes, represented by a dataset containing information about 45,603 records with students' certification assessment scores. This compiled dataset provides a rich resource for researchers studying digital literacy and learning analytics. It offers researchers the opportunity to gain valuable insights, inform evidence-based educational practices, and contribute to the ongoing efforts to improve digital literacy education in Thailand and beyond.

**Dataset:** https://dx.doi.org/10.21227/370s-1s37

**Dataset License:** CC-BY 4.0

**Keywords:** digital literacy dataset; IC3 certification; improvement; learning analytics; RMUTT



## 1. Summary

Digital literacy is a personal skill regarding one's ability to use a present digital technology for daily use, which includes operating, understanding, accessing, communicating, searching, and processing information technology [1]. In the 21st century, this set of skills and competencies is very important for professional life, Industry 4.0, and work in academic fields [2]. Nowadays, digital technology consists of hardware, software, and information. The technology can include personal computers, mobile phones, tablets, computer programs, and online media. Digital literacy is the primary factor affecting quality of life in the digital age. If a country fails to adopt and utilize information and communication technologies (ICTs), it will encounter digital exclusion as it cannot access conventional mainstream information sources [3].

The policy of Thailand 4.0 considers the country's economic development, providing a model for the development of the national economy by relying on the production structure and the occupational basis of people in Thai society [4]. Also, according to such policy, youth groups and students play an important role in the development of the country, as the youth population is three times greater than the working-age population. However, the main problem is that the ICT skill levels of the youth are rather low; this is a factor

that greatly influences the upgrading of the Thailand 4.0 policy. Similarly, the digital transformation process still encounters problems in many areas, and it is necessary for the population to develop fundamental digital skills to make the digital transformation process more efficient [5].

The Rajamangala University of Technology Thanyaburi (RMUTT) aligns itself with the vision of Thailand 4.0 and places great importance on students acquiring digital literacy skills. In line with this policy, RMUTT strives for a high success rate, aiming for nearly one hundred percent proficiency in digital literacy skills among the student population. RMUTT students have high expectations of the university, envisioning an educational environment that fosters excellence in the 21st century, with a particular emphasis on the development of digital literacy skills [6]. The IC3 Digital Literacy Certification (IC3) is a globally recognized standard utilized to certify individuals at entry and employee levels with sufficient ICT skills. In Thailand, the IC3 certification is widely adopted as a measure of digital literacy proficiency [7]. During the pilot phase, the IC3 certification was implemented at RMUTT as a testing standard as part of a short-term program aimed at enhancing students' digital skills. Although the percentage of passing examinations was at an acceptable level, the number of students participating in the program was still very small compared to the total number of students at the university.

To address this issue and expand student involvement, the university introduced a new general education subject relating to digital literacy, titled "Computer and Information Technology Skill" (RMUTT CITS course), during the first phase of the program in 2019. This initiative aims to increase the number of students engaging with and acquiring essential digital skills. Students from all faculties can register for this subject freely, and they also use IC3 as a testing standard in mid-year and final examinations. Moreover, the RMUTT Learning Management System (LMS) was employed as the primary platform used for learning this course to develop the ICT skills of students and lecturers with regard to using a digital platform. This LMS is not provided for full self-learning. It is used for learning activities such as online assignment submission and the provision of online resources. However, despite these efforts, the students' IC3 pass rate remained disappointingly low. This study investigated what factors influence students' digital skills and how we can elucidate the relationships among these factors.

From primary to higher education, the LMS has been utilized for years to facilitate the establishment of a good learning environment. With the rapid advancement of information technology, large-scale data collection on student populations is feasible. Several scientific researchers have studied the influence of student data analysis in recent years. This demonstrates the significance of open datasets, which provide a consistent method for comparing and visualizing results. There are several publicly accessible data sets discussed in previous studies. Table 1 showcases the dataset's contents, encompassing demographic information, academic records, and results from ICT skill tests. In contrast, the RMUTT Digital Literacy Dataset (RMUTT-DLD) offers a broader scope by including data on RMUTT students from 2016 to 2023. This extended dataset encompasses demographic information, academic learning records, and certification outcomes, providing a more comprehensive view of students' digital literacy progression over time.

Education has a substantial impact on economic growth and employment prospects. With the aim of providing students with the best learning resources, an abundance of predictive analytical educational research articles has been released in recent years. Over the past several years, effective statistical and machine learning approaches have been widely applied to educational datasets. For example, high school and college dropout rate datasets have been proposed by several researchers [8–10]. These datasets can be used to develop a model for predicting the dropout rate, which in turn may allow for the dropout rate to be lowered if the needs of students are better met. There was also a study that investigated the student dropout rate at the University Faculty of Electrical and Computer Engineering (FECE) from 2001 to 2015 [11]. This is why decreasing the number of students who drop out before graduating is so crucial. Using data mining techniques, [12] suggested

a novel recommendation system based on student data aimed at enhancing the number of university graduates by offering suitable subject selections.

In addition, higher-education students are continuously expected to improve their ICT competencies amongst the rapid development of the digital technology era. In 2017, [13] proposed a dataset that includes data from 22 courses presented by 32,593 Open University students (OU). The dataset contains demographic information as well as clickstream data gathered from student interactions in a virtual learning environment (VLE). In order to assess the impact of a VLE on learning outcomes, the VLE dataset was proposed. Some studies [11,12,14] have suggested datasets containing observations of students' ICT skill usage and evaluations of students' new technology learning skills. Digital Kids Asia Pacific (DKAP) published a new dataset encompassing 1061 observations of students' information and communication technology competence rates from several high schools across five Vietnamese regions and cities. The dataset includes responses from thousands of students who were asked to rate their digital literacy. Consequently, in order to address and analyze the university's digital literacy and provide the best quality education, our dataset, spanning from 2016 to 2023, consists of three main sections concerning the students' demographics, academic records, and IC3 digital literacy exam results.

**Table 1.** Comparison of recent datasets in the academic area.

| Dataset | Year | High School | Undergraduate | Number of Observations | Purpose | Location |
|---|---|---|---|---|---|---|
| Open University Learning Analytics Dataset [13] | 2017 | - | ✓ | 22 courses, 32,593 students | Students' interactions in the virtual learning environment (VLE) | Open University (OU) |
| Digital Competency Observation Dataset [15] | 2019 | ✓ | - | 1061 students | Digital competency | Vietnam |
| Academic Performance Evaluation Dataset [11] | 2020 | ✓ | ✓ | 12,411 students | Observe the influence of social variables and the evolution of students' learning skills | Colombia |
| Video Conferencing Tools Acceptance Dataset [14] | 2020 | - | ✓ | 277 records | Video conferencing tools (VTCs) | Vietnam |
| High-School Dropout Rate Dataset [10] | 2022 | ✓ | - | 1613 records | Student Dropout rates | United States |
| C# Programming Examination Dataset [12] | 2022 | - | ✓ | Unspecified | Academic results in C# programming language | Iraq, Sudan, Nigeria, South Africa, and India |
| Undergraduate and High-School Dropout Rate Dataset [9] | 2022 | ✓ | ✓ | 50 records, 143,326 records | Student dropout rate | Mexico |
| * RMUTT-DLD | 2023 | - | ✓ | 45,603 records | IC3 Digital Literacy Certification | Thailand |

Note: * The dataset in this study is called the RMUTT Digital Literacy Dataset (RMUTT-DLD).

## 2. Data Description

To fully comprehend the proposed dataset, a description of the RMUTT digital literacy learning process must be provided. RMUTT is one of Thailand's public universities, with approximately 25,000 students enrolled in various programs. The RMUTT LMS is used to deliver digital literacy-related learning resources. The database stores instructor and student interactions with course materials and assignments. It allows for the frequency of online assignment submissions in related modules to be lowest, low, medium, or high.

Students are aware of the policies regarding data protection and the ethics code in the use of student data recorded in databases for learning and research analysis. They are provided with crucial details on how their data is used and the possibility of data sharing with other academics for research purposes that can be disclosed to students. Additionally, this RMUTT-DLD dataset has been anonymized and cannot be used to identify specific pupils and lecturers.

This dataset comprises two distinct learning process application periods. The first term ran from 2016 to 2018, and the second from 2019 to 2023. Figure 1 depicts the learning process approach for the first period, in which all students studying the RMUTT CITS course had to register for a schedule of IC3 certification exams to evaluate their digital literacy skills. After receiving the schedule, students took the certification exam and received a score which equates to either a "Fail" or "Pass" grade. Also, the scores from such examinations were partially used for grading the course.

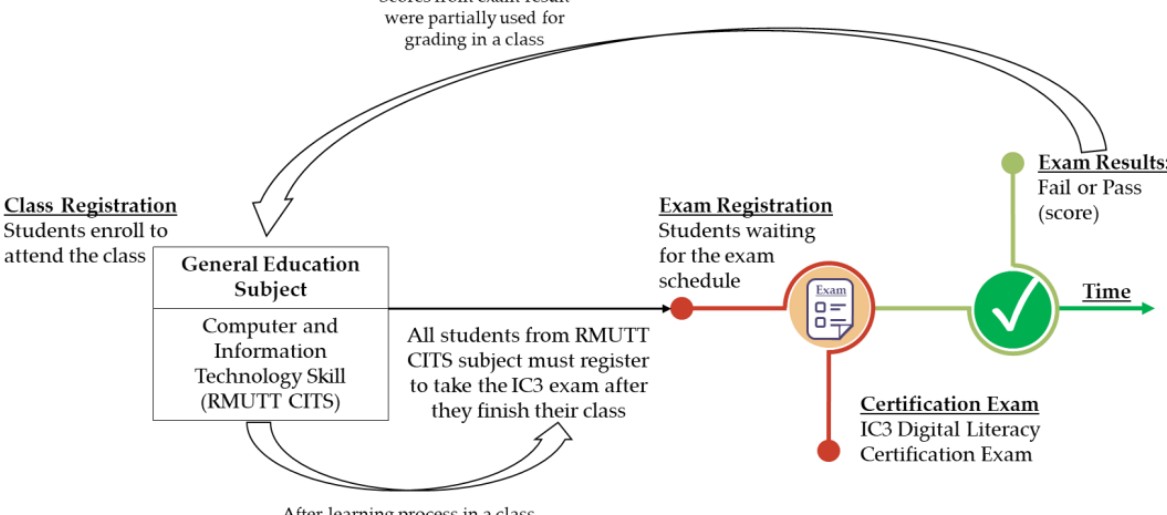

**Figure 1.** Digital literacy learning procedure—first period (2016–2018).

Figure 2 demonstrates the learning procedure for the second phase. As mentioned in the previous section, the initial phase of implementation did not achieve the desired outcomes. Therefore, RMUTT created a digital literacy improvement platform, including two modules. First, the self-e-Learning module was designed based on the standard gamification concept, and learners can study using that module completely on their own. Second, an intensive tutoring module was provided for a certain period. Typically, any student can register for the self-e-Learning module without registering for the RMUTT CITS course. For students who meet the qualification criteria, there is the option to participate in the intensive tutoring module. Additionally, students who are deemed qualified by the board of lecturers from the RMUTT CITS course can also directly access the intensive tutoring module. Then, students can take the IC3 certification exam in the first period.

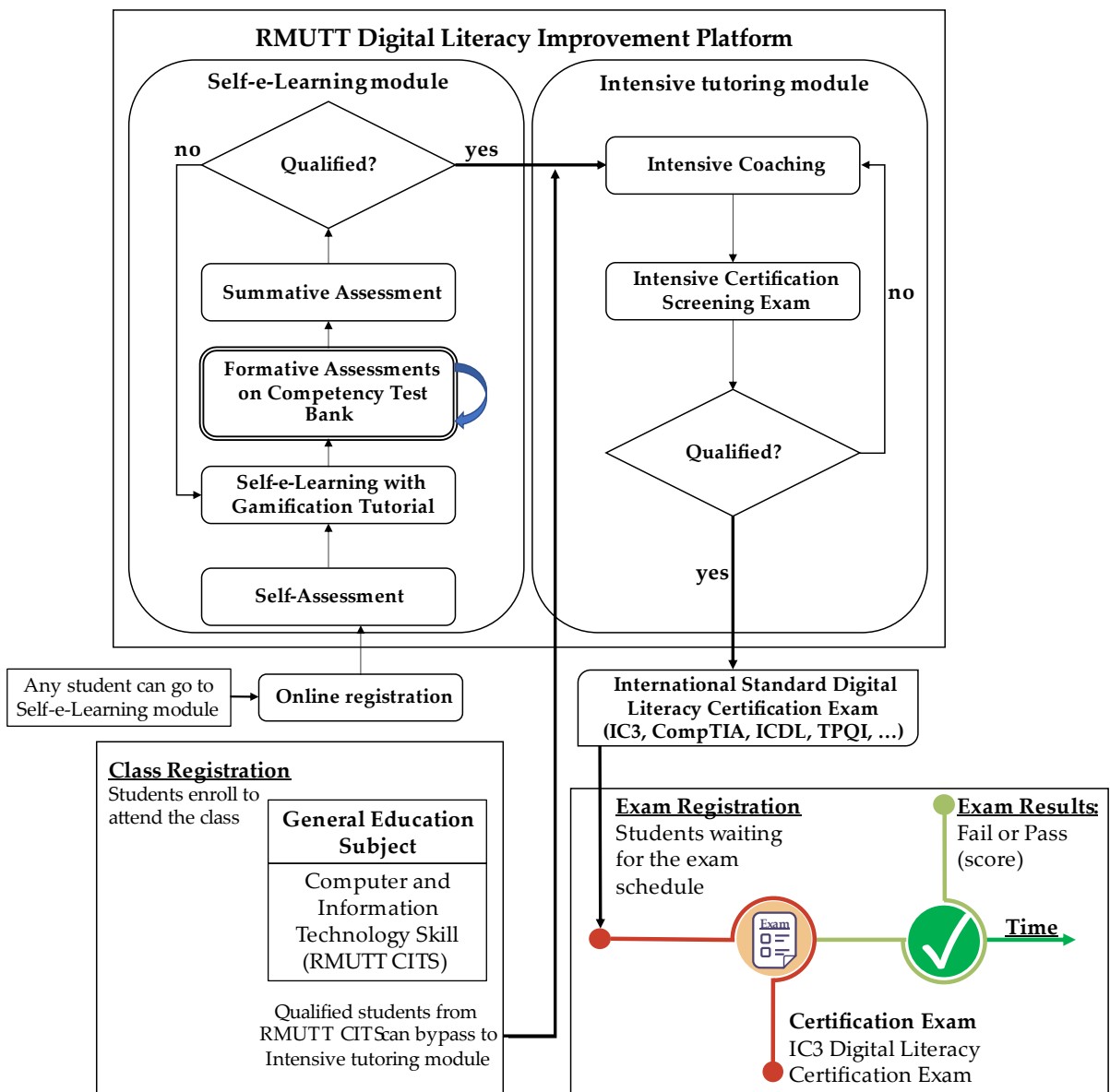

**Figure 2.** Digital literation learning procedure—second period (2019–2023).

Table 2 shows the detailed structure of the RMUTT-DLD dataset, consisting of the field name, data type, description, and data scope. The dataset is a collection of anonymous students' profiles, academic records, and IC3 digital literacy exam results, spanning from 2016 through 2023, as shown in Figure 3. The dataset focuses on students; hence, students are the focal point. Each record within the data corresponds to a student who registered for the IC3 certification exam in a specific module. The dataset includes a variety of demographic information, consisting of the student's encoded identifier, first-entry GPA, current GPA, admission year, faculty name in Thai and English, department name in Thai and English, home province name in Thai, home district name in Thai, and contact zip code in Thailand. The prefix "STD" was added to the field names of these data. The records of the IC3 exam results were combined with the students' profiles, which can refer to other fields, and the prefix "IC3" was added. The IC3 exam has three main modules, including 'IC3 GS5—Computing Fundamentals', 'IC3 GS5—Key Applications', and 'IC3 GS5—Living Online'. Information regarding the language, score, result, used time, station, and year of the IC3 examinations were also recorded. Furthermore, there are six fields of academic records, including the class identifier, teacher's encoded identifier in each class, number of

students who enrolled in a class, year of class opening, semester period, and frequency of online assignment submissions. The prefixes "CLASS" and "ONLINE" were added to the academic records.

**Table 2.** The detailed structure of the RMUTT-DLD dataset.

| No. | Field Name | Data Type | Description | Data Scope |
|---|---|---|---|---|
| 1 | STD_ENCODE_ID | Text | Record of student's encoded identifier. | There are 45,603 IC3 examination records that were recorded. |
| 2 | IC3_MODULE_NAME | Text | IC3 certificate module name. This field has only three modules. | IC3 GS5—Computing Fundamentals IC3 GS5—Key Applications IC3 GS5—Living Online |
| 3 | IC3_EXAM_LANGUAGE | Text | Language for examination. | English/ Thai. |
| 4 | IC3_SCORE | Integer | IC3 certificate score for each module. | 0 to 1000 points. |
| 5 | IC3_RESULT | Text | IC3 certificate result. Scores $\geq$ 700 pass; otherwise, fail. | Fail/ Pass. |
| 6 | IC3_EXAM_TIMEUSED | Integer | The time that was used during the examination. | 0 to 3000 s. |
| 7 | IC3_EXAM_STATION | Text | Station of the test taker, mostly including building and computer name. For example, IWORK-201-01 is IWORK building, room number 201, and computer number 01. | There are 997 stations. Some are not in the standard format because they may use an extra building or computer. |
| 8 | IC3_EXAM_YEAR | DateTime (Year) | Year of IC3 examination in yyyy format, such as 2023. | 2016 to 2023 A.D. |
| 9 | STD_ENTRY_GPA | Float | Student's first-entry GPA | 1.0 to 4.0 on a 4.0 scale. |
| 10 | STD_CURRENT_GPA | Float | Student's current GPA during the IC3 examination. | 0.0 to 4.0 on a 4.0 scale. |
| 11 | STD_ADMIT_YEAR | DateTime (Year) | Student's admission year in yyyy format, such as 2022. | 2012 to 2022 A.D. |
| 12 | STD_FACULTYNAME_THAI | Text | Student's faculty name in Thai. | There are 13 faculties. |
| 13 | STD_FACULTYNAME_ENG | Text | Student's faculty name in English. | There are 13 faculties. |
| 14 | STD_DEPARTMENTNAME_THAI | Text | Student's department name in Thai. | There are 43 departments. |
| 15 | STD_DEPARTMENTNAME_ENG | Text | Student's department name in English. | There are 43 departments. |
| 16 | STD_HOME_PROVINCENAME | Text (GEO) | Student's home province name in Thai. | There are 77 provinces in Thailand. |
| 17 | STD_HOME_DISTRICT | Text (GEO) | Student's home district name in Thai. | There are 988 districts. |
| 18 | STD_CONTACT_ZIPCODE | Text (GEO) | Student's contact zip code in Thailand. In general, some districts have the same contact zip code. | There are 855 contact zip codes. Some values are NA, which is undefined. |
| 19 | CLASS_ID | Text | Class identifier is used for classifying a class/section for RMUTT CITS. | There are 788 sections for the RMUTT CITS class. |
| 20 | CLASS_TEACHER_ENCODE_ID | Text | Record of teacher's encode identifier. This field can distinguish a lecturer from each other. | There are 76 teachers who taught many classes and have different name IDs. |
| 21 | CLASS_ENROLLSEAT | Integer | Number of students who enrolled in a class. | Between 3 and 78 students in a class. |
| 22 | CLASS_ACADEMIC_YEAR | DateTime (Year) | Year of class opening in yyyy format, such as 2022. | 2015 to 2022 A.D. |
| 23 | CLASS_SEMESTER | Integer | Semester period in which the class opens. | Semester 1, 2, or 3. |
| 24 | ONLINE_ASSIGNMENT _SUBMISSION_FREQUENCY | Text | Frequency of online assignment submissions in related modules. This field was transformed to include four levels. | Lowest/Low/Medium/High |

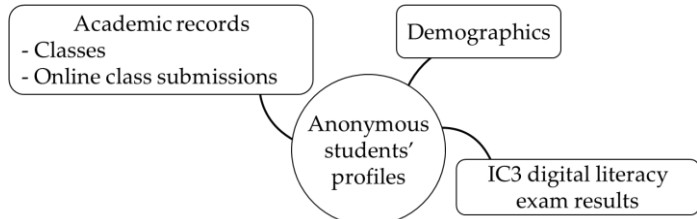

**Figure 3.** Overall dataset structure diagram.

The dataset is available in the .xlsx format and comprises three modules with 45,603 enrolled students. It can be freely downloaded by visiting the provided link via the file named "RMUTT-DLD-dataset-master.xlsx". This dataset can be imported into any application for further analysis or use, making it applicable to various scenarios. It facilitates the evaluation of predictive models to anticipate students' certification exam results and allows for model comparisons with those created by other researchers.

One interpretation of the dataset can be seen in Figures 4 and 5, and there are significant differences between the results of the certification exams before (2016–2018) and after (2019–2023) the digital literacy platform improvement. This is due to differences in the digital literacy learning procedure, which was explained in the previous paragraph. The differences are clear; the pass rate in 2019–2023 was better than the previous period.

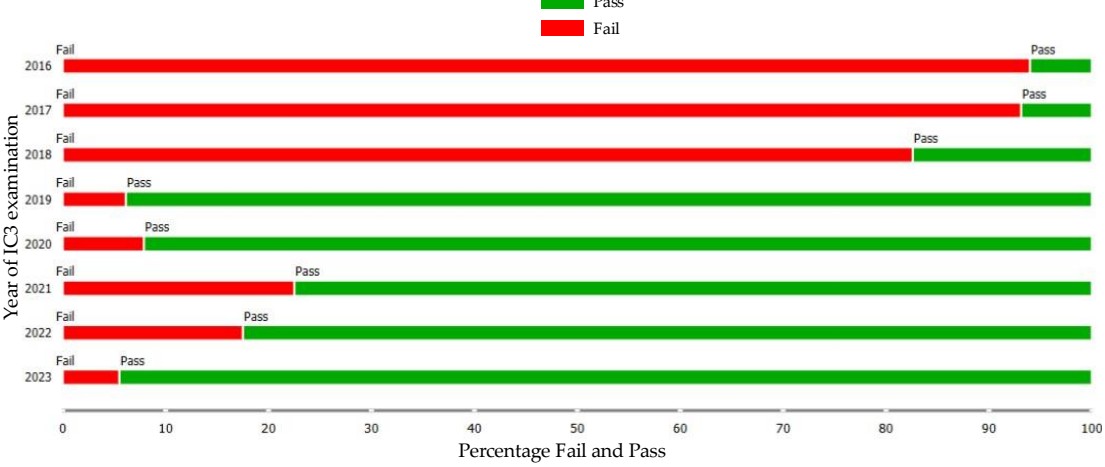

**Figure 4.** Certification exam results from 2016 to 2023 (percentage).

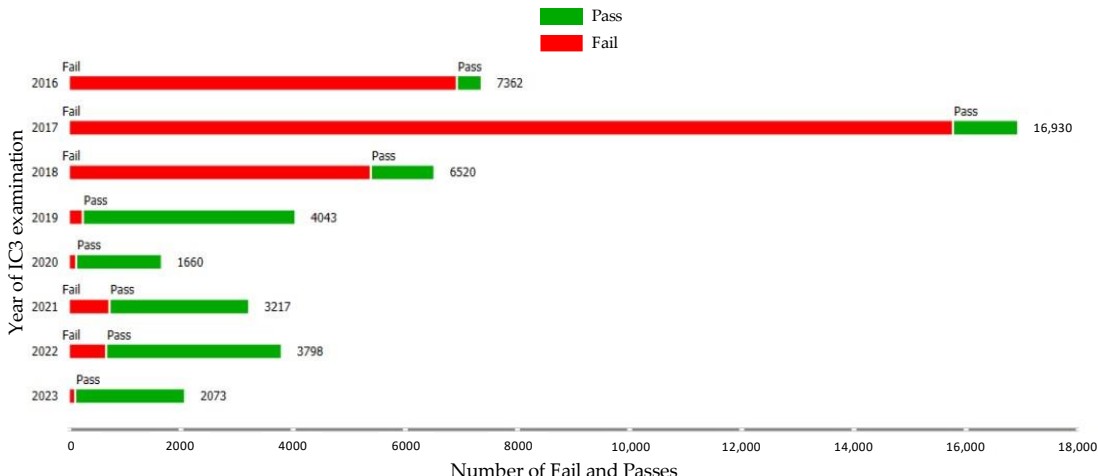

**Figure 5.** Certification exam results from 2016 to 2023 (number).

In Figure 6, an interpretation of the data is presented, showing the relationship between the number of assignments and the pass rate of students who took the IC3 exam. The data visualization divides the data into two categories: before and after the platform improvement. Furthermore, Figure 7 illustrates the distribution of the student population across Thailand and IC3 exam pass rates based on their province of residence. It is evident that students residing in the central Thailand area exhibited superior digital literacy skills compared to other provinces, on average.

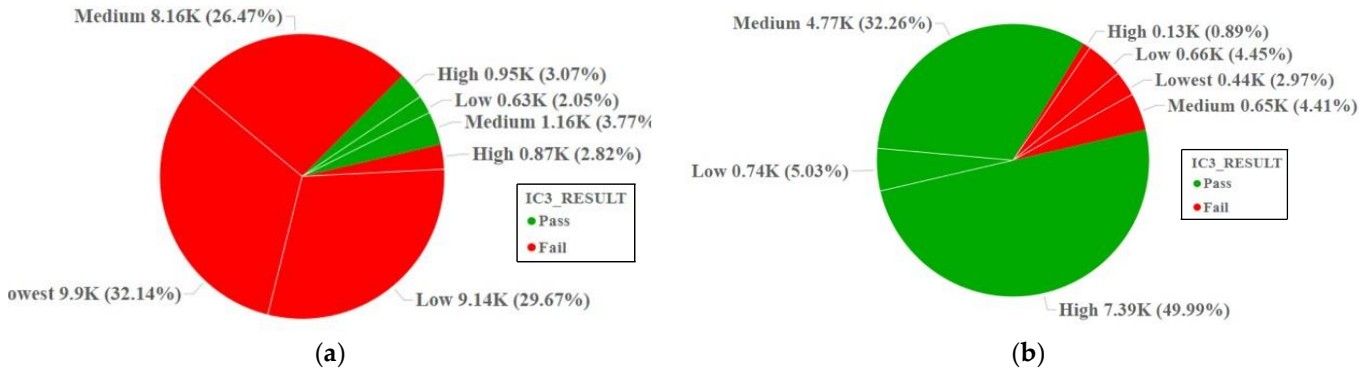

(a)                                                          (b)

**Figure 6.** Online assignment submission frequency and IC3 result rate. (**a**) Before improvement (2016–2018); (**b**) after improvement (2019–2023).

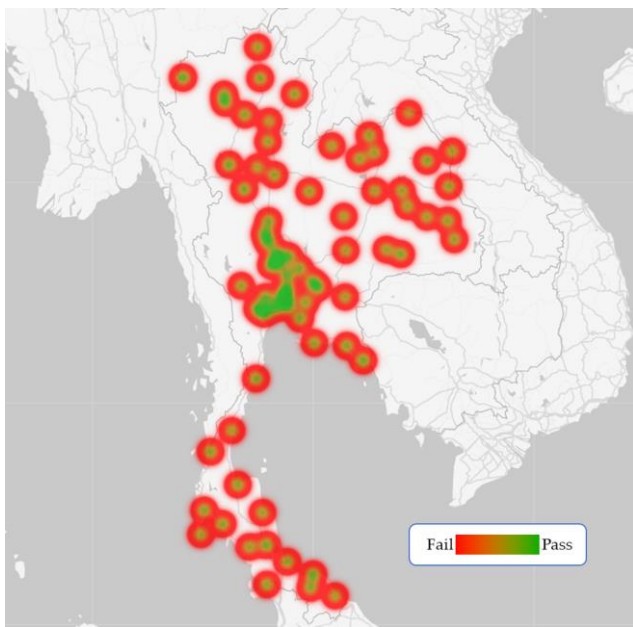

**Figure 7.** The demographic makeup of RMUTT students in the Thailand region.

## 3. Methods

### 3.1. Raw Data

The data preparation process involved three key stages: raw data handling, data cleansing, data anonymization, and release, as shown in Figure 8. The first stage was raw data handling, which encompassed the collection, extraction, and initial storage of data from various sources. Students at RMUTT have access to a variety of information system technologies that can be used to support their academic activities. As mentioned in the previous section, RMUTT has a data center for collecting all information due to the significant variation between information systems. In the dataset utilized for this article, three distinct types of data are distinguished:

- Demographic data—represent basic information on the students, such as name, age (date of birth), home province, home district, first-entry GPA, current GPA, faculty name, etc.
- Academic data—show the records of enrollment information of a student's education at RMUTT, including information on teachers, classes, and activities in RMUTT LMS.
- IC3 digital literacy exam data—are a record of student exam results according to digital literacy abilities.

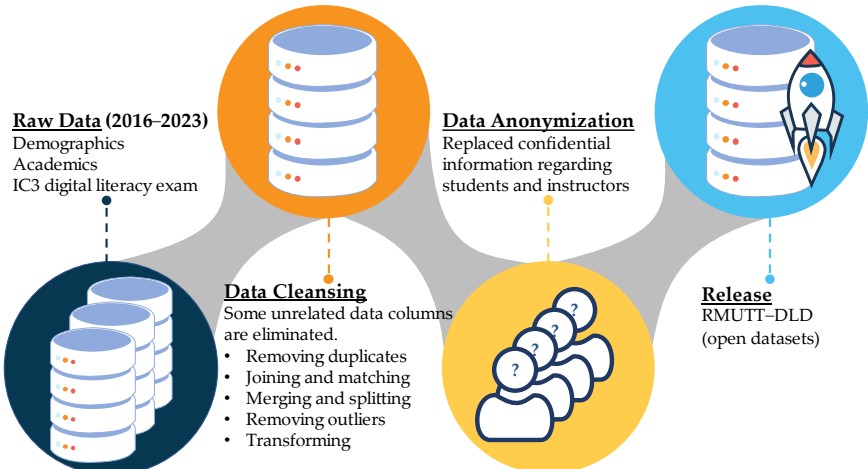

**Figure 8.** Dataset preparation process overview.

### 3.2. Data Cleansing

Data cleansing describes the activity of detecting and correcting mistaken records in a dataset. The data center has collected demographic, academic, and digital literacy exam data on students since 2016. We compiled information on digital literacy exams given at RMUTT between 2016 and 2023. Due to the records coming from various sources, they were combined with student ID, which can represent a specific source. Insignificant fields were also removed because there are some repeated values, such as payment type, voucher, and exam level. Some examples of data cleansing processes used in the study include:

- Removing the duplicated data and unused columns from the raw dataset.
- Joining, merging, and splitting the data among sources using student ID as a key.
- Removing outliers from data sources. For example, the minus values of GPA on a 4.0 scale were removed because the data were sometimes entered incorrectly from the beginning.
- Transforming some local data to international data units, such as year in B.E. into A.D. format, and the number of assignments submitted into the four simplified levels.

### 3.3. Data Anonymization and Release

The dataset anonymization procedure was built in accordance with RMUTT's ethical and privacy guidelines. The entire process of creating and releasing datasets is overseen by the RMUTT administration and approved by the Academic Resources and Information Technology (ARIT) departments. Self-anonymization is accomplished through a series of stages. The first step is to replace student and instructor personal information. This includes the student's ID number, instructor's name, and RMUTT-specific identification.

### 4. Data Evaluation

As a preliminary evaluation, a correlation matrix analysis was conducted on the dataset. This analysis is performed to identify relationships, explore data, select variables, and make data-driven decisions. The correlation matrix heatmap, as depicted in Figure 9, is a visual representation of the correlation values between different variables in a dataset. Each cell in the heatmap corresponds to the correlation coefficient between two variables.

The correlation coefficient ranges from negative one to one, indicating the strength and direction of the relationship between the variables. The correlation analysis (Figure 9) reveals several noteworthy findings concerning the relationships between different variables:

(1) The variables IC3_Score, IC3_Result, and IC3_Exam_Timeused exhibit a high correlation with each other, indicating that a negative correlation is observed between IC3_Exam_Timeused and performance, suggesting that students who take more time to complete the exam tend to have lower scores.

(2) Variables such as IC3_Exam_Year, Std_Admit_Year, Class_Id, and Class_Academic_Year demonstrate a positive correlation with IC3_Score and IC3_Result. This implies that students who enrolled after the implementation of the digital literation learning procedure achieved better scores and higher pass rates.

(3) Std_Entry_GPA and Std_Current_GPA also show a positive correlation with IC3_Score and IC3_Result. This suggests that students with strong entry and current GPAs tend to obtain higher IC3 scores and pass the exam.

(4) The variable Class_Teacher_Encoded_Id plays a role in determining IC3_Score and IC3_Result. This indicates that the selection of a teacher can influence a student's grades and overall success, as different teachers may vary in their delivery of course materials.

(5) The frequency of Online_Assignment_Submission is also correlated with IC3_Score and IC3_Result. A lower frequency of assignments given in a class is associated with lower scores and pass rates for students.

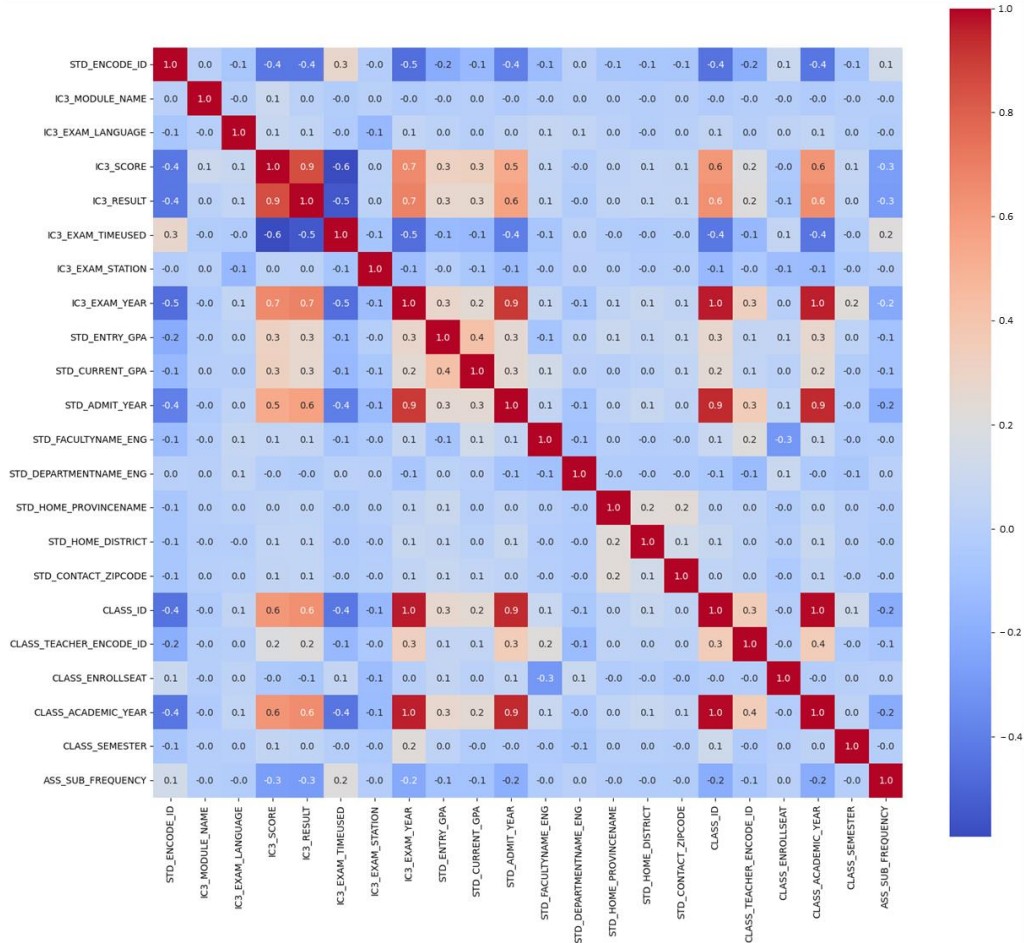

**Figure 9.** Dataset correlation matrix heatmap.

As a further method of evaluation, an open-source Orange [16] application was used to evaluate this dataset. Data may now be dynamically analyzed and more aesthetically visualized using Orange. Additionally, supported by this program are a number of ma-

chine learning methods that may be quickly and easily set up using a visual workflow. Figure 10 depicts the workflow used in this study. Six algorithms, Naïve Bayes [17], Logistic Regression [18], kNN [19], Random Forest [20], Support Vector Machine (SVM) [21], and Neural Network [22], were used to assess the accuracy of student certification results as predictors. Using a stratified tenfold cross-validation sample type with the average across classes as the target class, the data population was randomly chosen to be a sample dataset. Figure 11 shows the features and target used, based on the correlation analysis.

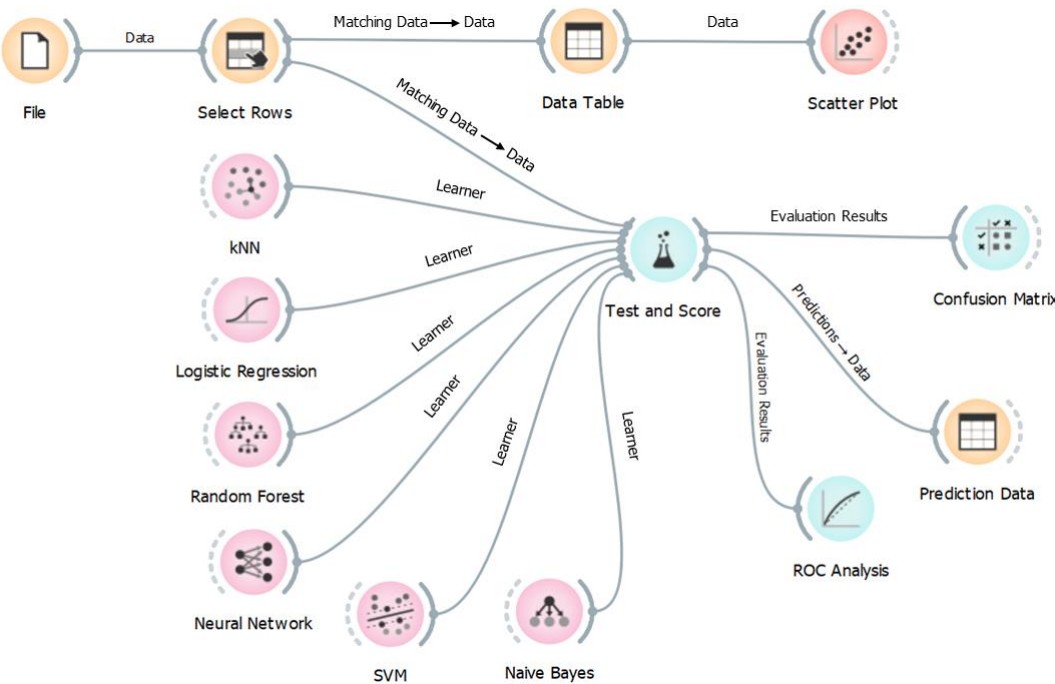

**Figure 10.** Classification model workflow using Orange.

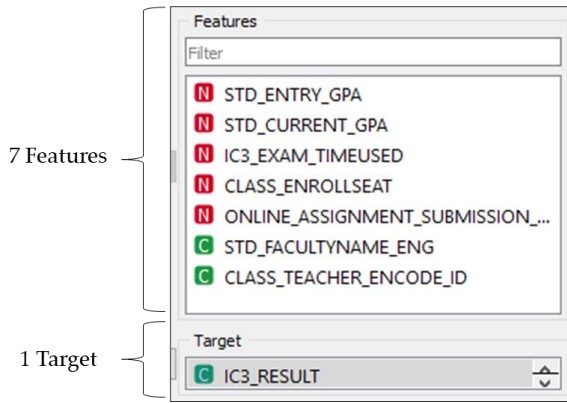

**Figure 11.** Dataset features used as evaluation.

Algorithm performance comparison can be seen through the Receiver Operating Characteristic (ROC) curve [23]. The evaluation results are then presented in the form of a confusion matrix based on Equations (1)–(6) regarding accuracy, true positive (*TP*) rate, false positive (*FP*) rate, recall, precision, and F1 measure [24]. Figure 12a–f shows each of the confusion matrices of the six algorithms used. A comparative evaluation of the six algorithms is presented in Table 3. Classification accuracy (CA), precision rate, area

under the ROC curve (AUC), F1 score, and recall were the metrics used to evaluate the data mining classifiers.

$$Accuracy = \frac{TP + TN}{TP + FP + TN + FN} \tag{1}$$

$$True\ positive\ rate = \frac{TP}{TP + FN} \tag{2}$$

$$False\ positive\ rate = \frac{FP}{FP + TN} \tag{3}$$

$$Recall = \frac{TP}{TP + FN} \tag{4}$$

$$Precision = \frac{TP}{TP + FP} \tag{5}$$

$$F_1\,Measure = \frac{2 \times Precision \times Recall}{Precision + Recall} \tag{6}$$

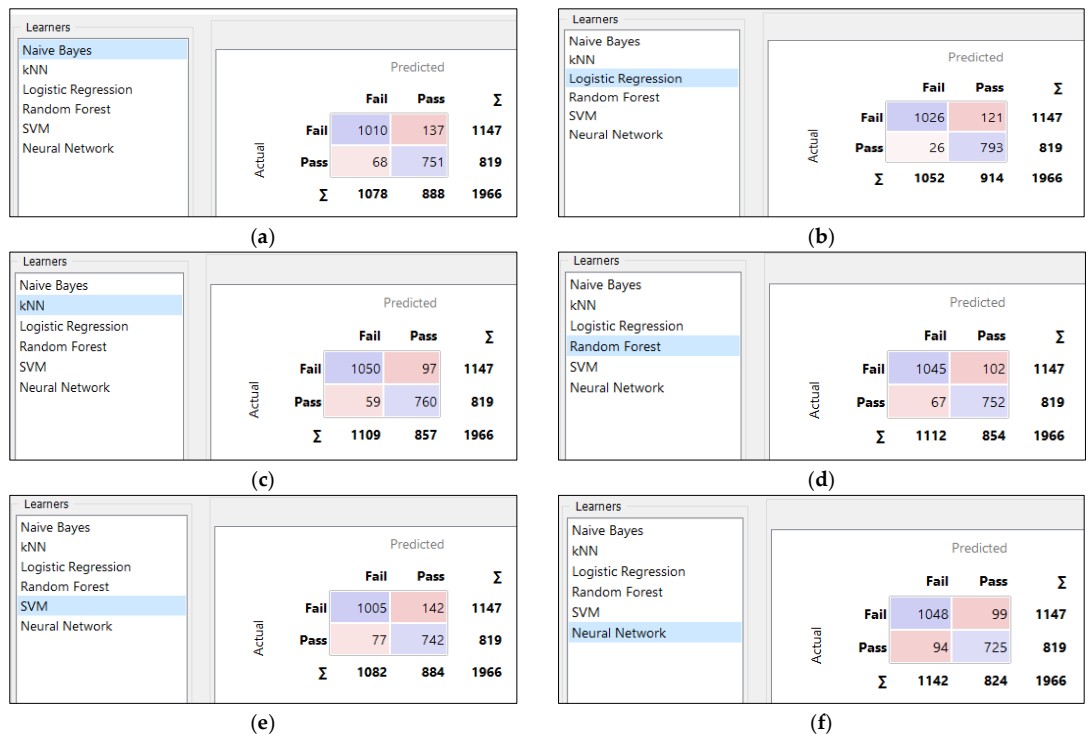

**Figure 12.** Confusion matrix of evaluation results. (**a**) Naïve Bayes. (**b**) Logistic Regression. (**c**) kNN. (**d**) Random Forest. (**e**) SVM. (**f**) Neural Network.

**Table 3.** Evaluation results.

| Model | AUC | CA | F1 | Precision | Recall |
|---|---|---|---|---|---|
| Logistic Regression | 0.976 | 0.925 | 0.926 | 0.930 | 0.925 |
| kNN | 0.976 | 0.921 | 0.921 | 0.922 | 0.921 |
| Random Forest | 0.974 | 0.914 | 0.914 | 0.915 | 0.914 |
| Neural Network | 0.974 | 0.902 | 0.902 | 0.902 | 0.902 |
| Naïve Bayes | 0.952 | 0.896 | 0.896 | 0.899 | 0.896 |
| SVM | 0.934 | 0.889 | 0.889 | 0.892 | 0.889 |

The ROC curve can be used to graphically assess the accuracy of predictions. Plotting the anticipated true positive (TP) rate against the predicted false positive (FP) rate as a

gauge of the effectiveness of the classification algorithm led to the creation of the ROC curve. Figure 13a,b presents the ROC curve for the prediction analysis of pass and fail student certification scores, illustrating the differences in the predictive performance of the six methods. As the final stage of evaluation, visualization of the data was carried out into a scatter plot so that the data could be read more easily. Figure 14 shows the relationship between the IC3 score and IC3 exam time used, illustrating that students who spend more time tend to have lower scores, as mentioned in the correlation matrix heatmap. Meanwhile, Figure 15 shows the relationship between the faculties and the results of the IC3 certification result, where almost all students from the faculty of Fine and Applied Arts experience failure. This is because the majority of the education provided by this faculty is not primarily related to basics of ICT skills. Moreover, Figure 16 shows the relationship between the teachers who teach the course and the IC3 certification result. This means that teachers also affect the students' experience of failure or success.

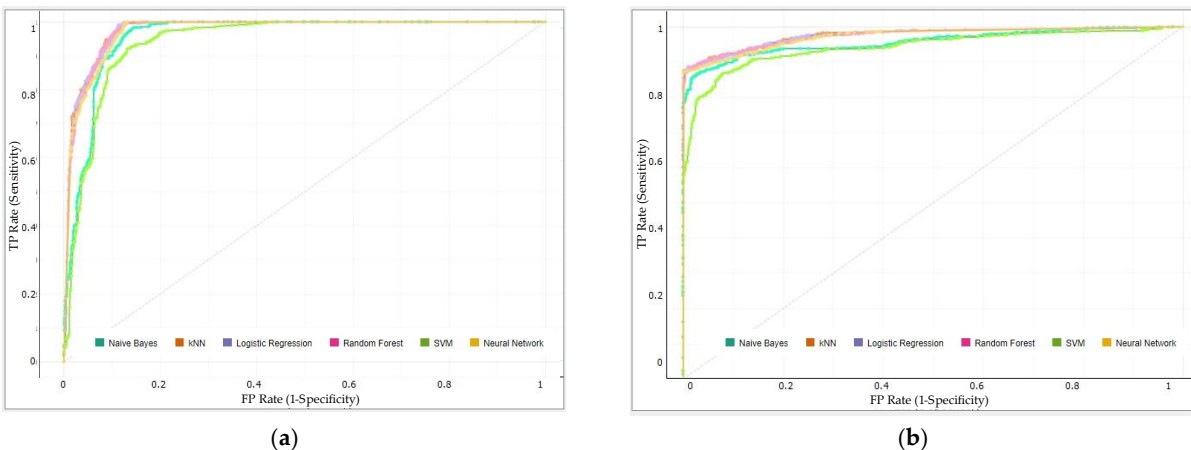

**Figure 13.** ROC curve. (**a**) Fail. (**b**) Pass.

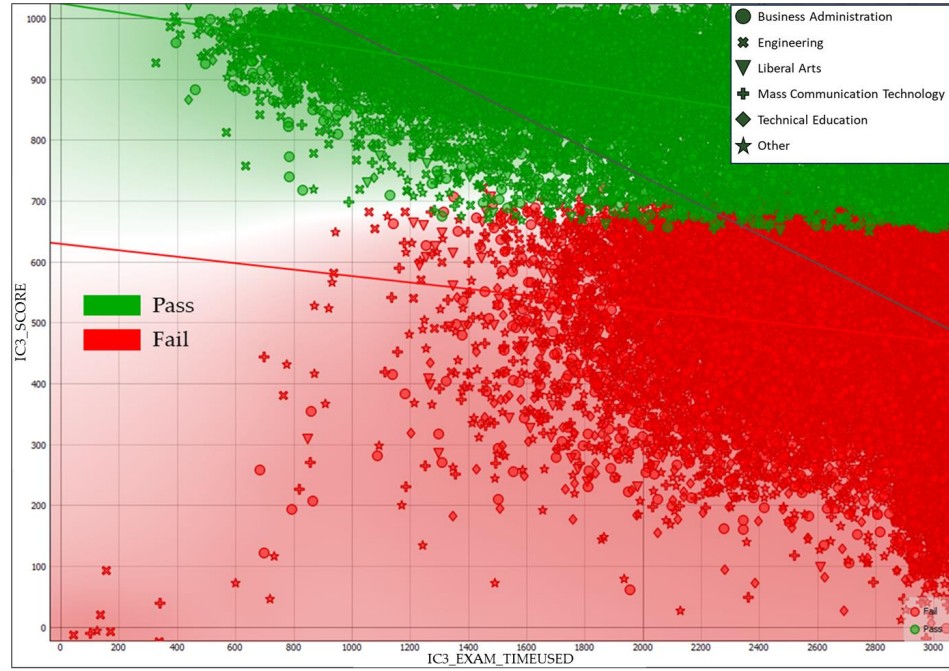

**Figure 14.** Scatter plot of IC3 score to IC3 exam time used.

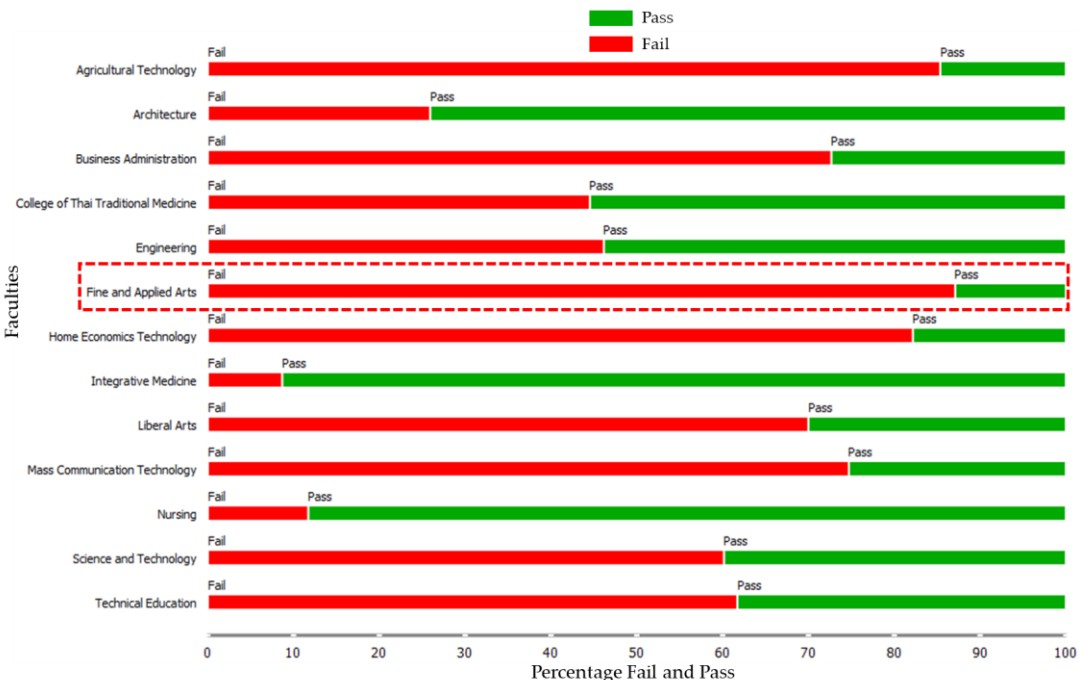

**Figure 15.** Certification exam results based on faculties (percentage).

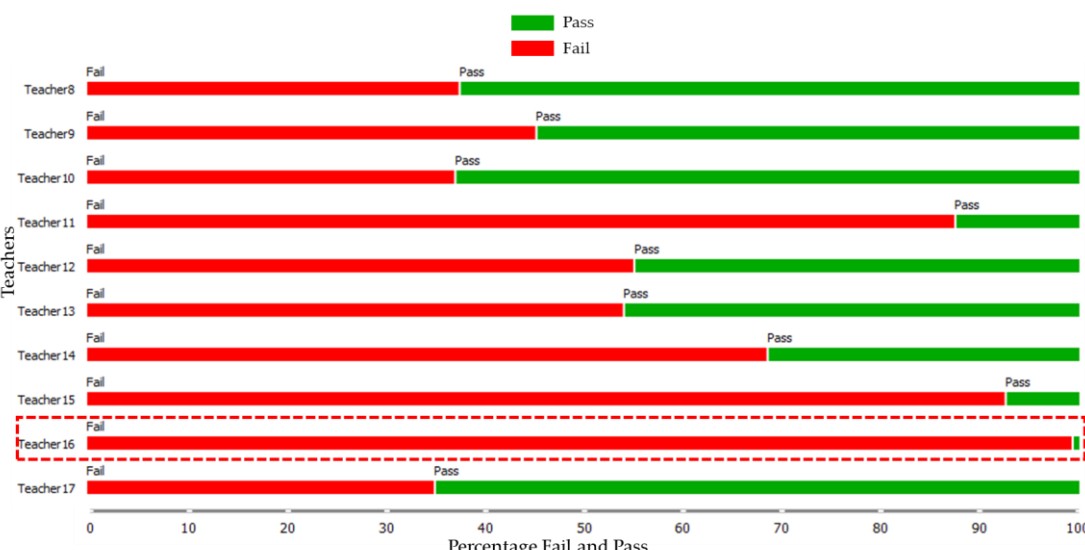

**Figure 16.** Certification exam results based on teachers (percentage).

## 5. Conclusions

This data descriptor presents a dataset created based on data obtained from the Rajamangala University of Technology Thanyaburi (RMUTT) called the RMUTT-DLD dataset, including the collection methodology for data preparation. This dataset is an amalgamation of several separate databases related to IC3 digital literacy certification results for students enrolled in the RMUTT CITS course. This dataset contains 45,603 records with 24 main variables and was collected between 2016 and 2023, including students' profiles and demographics, academic records, and IC3 digital literacy exam results. Also, the digital literacy learning procedure used between 2016 and 2018 was changed to the new implementation for improvement used between 2019 and 2023. Evaluation of the dataset was carried out by applying six machine learning algorithms. Making the right model based on this dataset will benefit students by implementing the right strategy to support student certification pass rates, especially in the field of digital literacy. To predict

student/instructor performance and recognize pupils at risk of failing, new or improved models are required. In summary, the availability of the RMUTT-DLD dataset, along with the detailed methodology and evaluation results, presents numerous opportunities for teachers, universities, and researchers. It enables them to leverage the dataset for research, replicate the methodology for data collection in their own contexts, and gain insights to improve digital literacy programs and support student success. Furthermore, this dataset is useful for researchers who wish to conduct comparative studies on the performance of student digital literacy competencies and for training in the field of machine learning.

**Author Contributions:** Conceptualization, P.N. and P.C.; methodology, P.N. and P.C.; software, P.N. and P.C.; validation, P.N. and P.C.; formal analysis, P.N. and P.C.; investigation, P.N. and P.C.; resources, P.N. and P.C.; data curation, P.N. and P.C.; writing—original draft preparation, P.N.; writing—review and editing, P.N. and P.C.; visualization, P.N.; supervision, P.C.; project administration, P.C. All authors have read and agreed to the published version of the manuscript.

**Funding:** This research received no external funding.

**Institutional Review Board Statement:** Privacy issues related to the collection, curation, and publication of student data were validated with RMUTT Data Owners and the Academic Resources and Information Technology (ARIT) departments.

**Informed Consent Statement:** Not applicable.

**Data Availability Statement:** The data presented in this study are openly available at https://dx.doi.org/10.21227/370s-1s37 (accessed on 25 June 2023).

**Acknowledgments:** We would like to express our deepest gratitude to King Mongkut's Institute of Technology Ladkrabang (KMITL), Rajamangala University of Technology Thanyaburi (RMUTT), and Academic Resources and Information Technology RMUTT for the support and facilities that were provided for this research.

**Conflicts of Interest:** The authors declare no conflict of interest.

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
