# Peer review of "Knowledge Discovery and Dataset for the Improvement of Digital Literacy Skills in Undergraduate Students"

_data, 2023_

Round 1

Reviewer 1 Report

I have to congratulate the authors because their research was very complicated and I think that it can be very useful for a lot of future and current students.

The authors made very hard work to complete the data, analyze them, and made recommendations.

the paper is very interesting for readers and it is easy to understand, what is necessary for the students according to the actual respectively future skills that they have to know.

Recommendations for authors:

1. Figure 9 Dataset correlation matrix heatmap is a little bit unclear, please, specify it more.

2. Please, extend the conclusion.

3. All figures have poor quality - please improve the resolution of the figures.

I think that the paper fulfills the requirements for English, but there are only a few mistakes.

Reviewer 2 Report

The article seems to me to be very complete in terms of methodology and presentation of the data. I recommend expanding on how this information can be used by other teachers or universities, how this methodology can be replicated and what the benefits would be.
